# RotoGrad: Gradient Homogenization in Multi-Task Learning

## Abstract

Multi-task learning is being increasingly adopted in applications domains like computer vision and reinforcement learning. However, optimally exploiting its advantages remains a major challenge due to the effect of negative transfer. Previous works have tracked down this issue to the disparities in gradient magnitudes and directions across tasks, when optimizing the shared network parameters. While recent work has acknowledged that negative transfer is a two-fold problem, existing approaches fall short as they focus only on either homogenizing the gradient magnitude across tasks; or greedily change the gradient directions, overlooking future conflicts. In this work, we introduce RotoGrad, an algorithm that tackles negative transfer as a whole: it jointly homogenizes gradient magnitudes and directions, while ensuring training convergence. We show that RotoGrad outperforms competing methods in complex problems, including multi-label classification in CelebA and computer vision tasks in the NYUv2 dataset.

## 1 Introduction

As neural network architectures get larger in order to solve increasingly more complex tasks, the idea of jointly learning multiple tasks (for example, depth estimation and semantic segmentation in computer vision) with a single network is becoming more and more appealing. This is precisely the idea of multi-task learning (MTL) [3], which promises higher performance in the individual tasks and better generalization to unseen data, while drastically reducing the number of parameters [27].

Unfortunately, sharing parameters between tasks may also lead to difficulties during training as tasks compete for shared resources, often resulting in poorer results than solving individual tasks, a phenomenon known as *negative transfer* [27]. Previous works have tracked down this issue to the two types of differences between task gradients. First, *differences in magnitude* across tasks can make some tasks dominate the others during the learning process. Several methods have been proposed to homogenize gradient magnitudes such as MGDA [28], GradNorm [6], or IMTL-G [18]. However, little attention has been put towards the second source of the problem: *conflicting directions* of the gradients for different tasks. Due to the way gradients are added up, gradients of different tasks may cancel each other out if they point to opposite directions of the parameter space, thus leading to a poor update direction for a subset or even all tasks. Only very recently a handful of works have started to propose methods to mitigate the conflicting gradients problem, for example, by removing conflicting parts of the gradients [33], or randomly 'dropping' some elements of the gradient vector [7].

In this work we propose RotoGrad, an algorithm that tackles negative transfer as a whole by homogenizing both gradient magnitudes and directions across tasks. RotoGrad addresses the gradient magnitude discrepancies by re-weighting task gradients at each step of the learning, while encouraging learning those tasks that have converged the least thus far. In that way, it makes sure that no task is overlooked during training. Additionally, instead of directly modifying gradient directions, RotoGrad

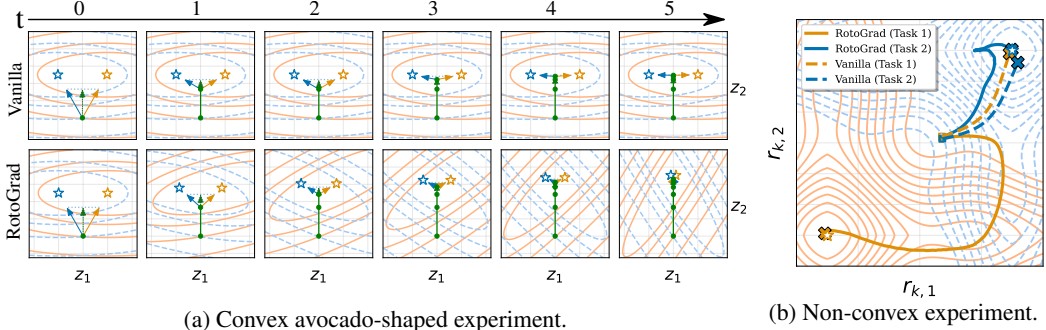

(a) Convex avocado-shaped experiment.

(b) Non-convex experiment.

Figure 1: Level plots showing the evolution of two regression MTL problems with/without RotoGrad, see Section 4. RotoGrad is able to reach the optimum (☆) for both tasks. *(a)* In the space of $\boldsymbol{z}$, RotoGrad rotates the function-spaces to align task gradients (blue/orange arrows), finding shared features $\boldsymbol{z}$ (green arrow) closer to the (matched) optima. *(b)* In the space of $\boldsymbol{r}_k$, RotoGrad rotates the shared feature $\boldsymbol{z}$, providing per-task features $\boldsymbol{r}_k$ that better fit each task.

smoothly rotates the shared feature space differently for each task, seamlessly aligning gradients in the long run. As shown by our theoretical insights, the cooperation between gradient magnitude- and direction-homogenization ensures the stability of the overall learning process. Finally, we run extensive experiments to empirically demonstrate that RotoGrad leads to stable (convergent) learning, scales up to complex network architectures, and outperforms competing methods in multi-label classification settings in CIFAR10 and CelebA, as well as in computer vision tasks using the NYUv2 dataset. Alongside this paper, we will provide a simple-to-use library to include RotoGrad in any Pytorch pipeline with a few lines of code.

## 2 Multi-task learning and negative transfer

The goal of MTL is to simultaneously learn $K$ different tasks, that is, finding $K$ mappings from a common input dataset $\boldsymbol{X} \in \mathbb{R}^{N \times D}$ to a task-specific set of labels $\boldsymbol{Y}_k \in \mathbb{Y}_k^N$. Most settings consider a hard-parameter sharing architecture, which is characterized by two components: the *backbone* and *heads* networks. The backbone uses a set of shared parameters, $\boldsymbol{\theta}$, to transform each input $\boldsymbol{x} \in \boldsymbol{X}$ into a shared intermediate representation $\boldsymbol{z} = f(\boldsymbol{x}; \boldsymbol{\theta}) \in \mathbb{R}^d$, where $d$ is the dimensionality of $\boldsymbol{z}$. Additionally, each task $k = 1, 2, \ldots, K$ has a head network $h_k$, with exclusive parameters $\boldsymbol{\phi}_k$, that takes this intermediate feature $\boldsymbol{z}$ and outputs the prediction $h_k(\boldsymbol{x}) = h_k(\boldsymbol{z}; \boldsymbol{\phi}_k)$ for the corresponding task. This architecture is illustrated in Figure 2, where we have added task-specific rotation matrices $\boldsymbol{R}_k$ that will be necessary for the proposed approach, RotoGrad. Note that the general architecture described above is equivalent to the one in Figure 2 when all rotations $\boldsymbol{R}_k$ correspond to identity matrices, such that $\boldsymbol{r}_k = \boldsymbol{z}$ for all $k$.

MTL aims to learn the architecture parameters $\boldsymbol{\theta}, \boldsymbol{\phi}_1, \boldsymbol{\phi}_2, \ldots, \boldsymbol{\phi}_K$ by simultaneously minimizing all task losses, that is, $L_k(h_k(\boldsymbol{x}), \boldsymbol{y}_k)$ for $k = 1, \ldots, K$. Although this is a priori a multi-objective optimization problem [28], in practice a single surrogate loss consisting of a linear combination of the task losses, $L = \sum_k \omega_k L_k$, is optimized. While this approach leads to a simpler optimization problem, it may also trigger *negative transfer* between tasks, hurting the overall

$$\boldsymbol{x} \xrightarrow{f_{\boldsymbol{\theta}}} \boldsymbol{z} \begin{array}{c} \xrightarrow{\boldsymbol{R}_1} \boldsymbol{r}_1 \xrightarrow{h_{\phi_1}} L_1(h_1(\boldsymbol{r}_1), \boldsymbol{y}_1) \\ - \boldsymbol{R}_2 \rightarrow \boldsymbol{r}_2 \xrightarrow{h_{\phi_2}} L_2(h_2(\boldsymbol{r}_2), \boldsymbol{y}_2) \\ \cdots \qquad\qquad \cdots \\ \xrightarrow{\boldsymbol{R}_K} \boldsymbol{r}_K \xrightarrow{h_{\phi_K}} L_K(h_K(\boldsymbol{r}_K), \boldsymbol{y}_K) \end{array}$$

Figure 2: Hard-parameter sharing architecture including the rotation matrices $\boldsymbol{R}_k$ of RotoGrad.

MTL performance due to an imbalanced competition among tasks for the shared parameters [27].

The negative transfer problem can be studied through the updates of the shared parameters $\boldsymbol{\theta}$. At each training step, $\boldsymbol{\theta}$ is updated according to a linear combination of task gradients, $\nabla_{\boldsymbol{\theta}} L = \sum_k \omega_k \nabla_{\boldsymbol{\theta}} L_k$, which may suffer from two problems. First, **magnitude differences** of the gradients across tasks may lead to a subset of tasks dominating the total gradient, and therefore to the model prioritizing them over the others. Second, **conflicting directions** of the gradients across tasks may lead to update

directions that do not improve any of the tasks. Figure 1 shows an example of poor direction updates (left) as well as magnitude dominance (right).

In this work, we tackle negative transfer as a whole by homogenizing tasks gradients both in magnitude and direction. Note that homogenizing gradients with respect to $\boldsymbol{\theta}$ is equivalent to homogenizing gradients with respect to the shared feature $\boldsymbol{z}$ due to the chain rule, $\nabla_{\boldsymbol{\theta}} L_k = \nabla_{\boldsymbol{\theta}} \boldsymbol{z} \cdot \nabla_{\boldsymbol{z}} L_k$. Thus, from now on we focus on homogenizing the feature-level task gradients $\nabla_{\boldsymbol{z}} L_k$.

# 3   RotoGrad

In this section we introduce RotoGrad, a novel algorithm that addresses the negative transfer problem as a whole. RotoGrad consists of two building blocks which, respectively, homogenize task-gradient magnitudes and directions. Moreover, these blocks complement each other and provide convergence guarantees of the network training. Next, we detail each of these building blocks and show how they are combined towards an effective MTL learning process.

## 3.1   Gradient-magnitude homogenization

As discussed in Section 2, we aim to homogenize gradient magnitudes across tasks, as large magnitude disparities can lead to a subset of tasks dominating the learning process. Thus, the first goal of RotoGrad is to homogenize the magnitude of the gradients across tasks at each step of the training.

Let us denote the feature-level task gradient of the $k$-th task for the $n$-th datapoint, at iteration $t$, by $\boldsymbol{g}_{n,k} \coloneqq \nabla_{\boldsymbol{z}} L_k(h_k(\boldsymbol{x}_n), \boldsymbol{y}_{n,k})$, and its batch versions by $\boldsymbol{G}_k^\top \coloneqq [\boldsymbol{g}_{1,k}, \boldsymbol{g}_{2,k}, \ldots, \boldsymbol{g}_{B,k}]$, where $B$ is the batch size. Then, equalizing gradient magnitudes amounts to finding weights $\omega_k$ that normalize and scale each gradient $\boldsymbol{G}_k$, that is,

$$||\omega_k \boldsymbol{G}_k|| = ||\omega_i \boldsymbol{G}_i|| \quad \forall i \iff \omega_k \boldsymbol{G}_k = \frac{C}{||\boldsymbol{G}_k||} \boldsymbol{G}_k = C \boldsymbol{U}_k \quad \forall k, \tag{1}$$

where $\boldsymbol{U}_k \coloneqq \frac{\boldsymbol{G}_k}{||\boldsymbol{G}_k||}$ denotes the normalized task gradient and $C$ is the target magnitude for all tasks. Note that, in the above expression, $C$ is a free parameter that we need to select.

In RotoGrad, we select $C$ such that all tasks converge at a similar rate. We motivate this choice by the fact that, by scaling all gradients, we change their individual step size, interfering with the convergence guarantees provided by their Lipschitz-smoothness (for an introduction to non-convex optimization see, for example, [25]). Therefore, we seek for the value of $C$ providing the best step-size for those tasks that have converged the least up to iteration $t$. Specifically, we set $C$ to be a convex combination of the task-wise gradient magnitudes, $C \coloneqq \sum_k \alpha_k ||\boldsymbol{G}_k||$, where the weights $\alpha_1, \alpha_2, \ldots, \alpha_K$ measure the relative convergence of each task and sum up to one, that is,

$$\alpha_k = \frac{||\boldsymbol{G}_k||/||\boldsymbol{G}_k^0||}{\sum_i ||\boldsymbol{G}_i||/||\boldsymbol{G}_i^0||}, \tag{2}$$

with $\boldsymbol{G}_k^0$ being the initial gradient of the $k$-th task, i.e., the gradient at iteration $t = 0$ of the training.

As a result, we obtain a (hyper)parameter-free approach that equalizes the gradient magnitude across tasks to encourage learning slow-converging tasks. Note that the resulting approach resembles Normalized Gradient Descent (NGD) [8] for single-task learning, which has been proved to quickly escape saddle points during optimization [24]. Thus, we expect a similar behavior for RotoGrad, where slow-converging tasks will force quick-converging tasks to escape from saddle points.

The resulting training algorithm may however diverge as a consequence of constantly oscillating between (slow-converging) tasks. For example, in scenarios where one task improves, there is always another task(s) that deteriorates. Fortunately, as shown in the following result (proof in Appendix A), such a phenomenon does not appear in the absence of conflicting gradients.

**Proposition 3.1.** *Let* $\boldsymbol{G}_1, \boldsymbol{G}_2, \ldots, \boldsymbol{G}_K$ *be the task gradients with respect to* $\boldsymbol{Z}$ *as defined above. If* $K = 2$*; or* $\cos\_sim(\boldsymbol{G}_i, \boldsymbol{G}_j) \geq 0$ *pairwise; then there exists a small-enough step size* $\varepsilon > 0$ *such that, for all tasks, we have that* $L_k(h_k(\boldsymbol{Z} - \varepsilon \cdot C \sum_k \boldsymbol{U}_k; \boldsymbol{\phi}_k); \boldsymbol{Y}_k) < L_k(h_k(\boldsymbol{Z}; \boldsymbol{\phi}_k); \boldsymbol{Y}_k)$.

In other words, Proposition 3.1 shows that, when gradients do not conflict in direction with each other, following the feature-level gradient $C \sum_k \boldsymbol{U}_k$ improves all (lower-bounded) task losses for

the given batch. This result, while restricted to the given batch and to the gradient with respect to the shared representation $\boldsymbol{Z}$, still provides useful insights in favor of having as *desideratum* of an efficient MTL pipeline the absence of conflicting gradients.

## 3.2 Gradient-direction homogenization

In the previous subsection, we have shown that avoiding conflicting gradients may not only be necessary to avoid negative transfer, but also to ensure the stability of the training. In this section we introduce the second building block of RotoGrad, an algorithm that homogenizes task-gradient directions. The main idea of this approach is to smoothly rotate the feature-space $\boldsymbol{z}$ in order to reduce the gradient conflict between tasks—in following iterations—of the training by bringing (local) optima for different tasks closer to each other (in the parameter space). As a result, it complements the previous magnitude-scaling approach and reduces the likelihood of the training to diverge.

In order to homogenize gradients, for each task $k = 1, \ldots, K$, RotoGrad introduces a matrix $\boldsymbol{R}_k$ so that, instead of optimizing $L_k(\boldsymbol{z})$ with $\boldsymbol{z}$ being the last shared representation, we optimize an equivalent loss function $L_k(\boldsymbol{R}_k \boldsymbol{z})$. As we are only interested in changing directions (not the gradient magnitudes), we choose $\boldsymbol{R}_k \in SO(d)$ to be a rotation matrix[1] leading to per-task representations $\boldsymbol{r}_k \coloneqq \boldsymbol{R}_k \boldsymbol{z}$. RotoGrad thus extends the standard MTL architecture by adding task-specific rotations before each head, as depicted in Figure 2.

Unlike all other network parameters, matrices $\boldsymbol{R}_k$ do not seek to reduce their task's loss. Instead, these additional parameters are optimized to reduce the direction conflict of the gradients across tasks. To this end, for each task we optimize $\boldsymbol{R}_k$ to maximize the batch-wise cosine similarity or, equivalently, to minimize

$$\mathcal{L}_{\text{rot}}^k \coloneqq -\sum_n \langle \boldsymbol{R}_k^\top \widetilde{\boldsymbol{g}}_{n,k}, \boldsymbol{v}_n \rangle, \tag{3}$$

where $\widetilde{\boldsymbol{g}}_{n,k} \coloneqq \nabla_{\boldsymbol{r}_k} L_k(h_k(\boldsymbol{x}_n), \boldsymbol{y}_{n,k}))$ (which holds that $\boldsymbol{g}_{n,k} = \boldsymbol{R}_k^\top \widetilde{\boldsymbol{g}}_{n,k}$) and $\boldsymbol{v}_n$ is the target vector that we want all task gradients pointing towards. We set the target vector $\boldsymbol{v}_n$ to be the gradient we would have followed if all task gradients weighted the same, that is, $\boldsymbol{v}_n \coloneqq \frac{1}{K} \sum_k \boldsymbol{u}_{n,k}$, where $\boldsymbol{u}_{n,k}$ is a row vector of the normalized batch gradient matrix $\boldsymbol{U}_k$, as defined before.

As a result, in each training step of RotoGrad we simultaneously optimize the following two problems:

$$\mathcal{N}\text{etwork:} \underset{\boldsymbol{\theta}, \{\boldsymbol{\phi}\}_k}{\text{minimize}} \sum_k \omega_k L_k., \qquad \mathcal{R}\text{otation:} \underset{\{\boldsymbol{R}_k\}_k}{\text{minimize}} \sum_k \mathcal{L}_{\text{rot}}^k \tag{4}$$

The above problem can be interpreted as a Stackelberg game: a two player-game in which *leader* and *follower* alternately make moves in order to minimize their respective losses, $L_l$ and $L_f$, and the leader knows what will be the follower's response to their moves. Such an interpretation allows us to derive simple guidelines to guarantee training convergence—that is, that the network loss does not oscillate as a result of optimizing the two different objectives in Equation 4. Specifically, following Fiez et al. [10], we can ensure that problem 4 converges as long as the rotations' optimizer (leader) is a slow-learner compared with the network optimizer (follower). That is, as long as we make the rotations' learning rate decrease faster than that of the network, we know that RotoGrad will converge to a local optimum for both objectives. A more extensive discussion can be found in Appendix B.

## 3.3 RotoGrad: the full picture

After the two main building blocks of RotoGrad, we can now summarize the overall proposed approach in Algorithm 1. At each step, RotoGrad first homogenizes the gradient magnitudes such that there is no dominant task and the step size is set by the slow-converging tasks. Additionally, RotoGrad smoothly updates the rotation matrices—using the local information given by the task gradients—to seamlessly align task gradients in the following steps, thus reducing direction conflicts.

## 3.4 Practical considerations

In this section, we discuss the main practical considerations to account for when implementing RotoGrad and propose efficient solutions.

---

[1]The special orthogonal group, $SO(d)$, denotes the set of all (proper) rotation matrices of dimension $d$.

**Algorithm 1** Training step with RotoGrad

**Input** input samples $\boldsymbol{X}$, task labels $\{\boldsymbol{Y}_k\}$, network's (RotoGrad's) learning rate $\eta$ ($\eta_{\text{roto}}$)
**Output** backbone (heads) parameters $\boldsymbol{\theta}$ ($\{\boldsymbol{\phi}_k\}$), RotoGrad's parameters $\{\boldsymbol{R}_k\}$

1: compute shared feature $\boldsymbol{Z} = f(\boldsymbol{X}; \boldsymbol{\theta})$
2: **for** $k = 1, 2, \ldots, K$ **do**
3:     compute task-specific loss $L_k = \sum_n L_k(h_k(\boldsymbol{R}_k \boldsymbol{z}_n; \boldsymbol{\phi}_k), \boldsymbol{y}_{n,k})$
4:     compute gradient of shared feature $\boldsymbol{G}_k = \nabla_{\boldsymbol{z}} L_k$
5:     compute gradient of task-specific feature $\widetilde{\boldsymbol{G}}_k = \boldsymbol{R}_k \boldsymbol{G}_k$          ▷ Treated as constant w.r.t. $\boldsymbol{R}_k$.
6:     compute unitary gradients $\boldsymbol{U}_k = \boldsymbol{G}_k / ||\boldsymbol{G}_k||$
7:     compute relative task convergence $\alpha_k = ||\boldsymbol{G}_k|| / ||\boldsymbol{G}_k^0||$
8: **end for**
9: make $\{\alpha_k\}$ sum up to one $[\alpha_1, \alpha_2, \ldots, \alpha_K] = [\alpha_1, \alpha_2, \ldots, \alpha_K] / \sum_k \alpha_k$
10: compute shared magnitude $C = \sum_k \alpha_k ||\boldsymbol{G}_k||$
11: update backbone parameters $\boldsymbol{\theta} = \boldsymbol{\theta} - \eta C \sum_k \boldsymbol{U}_k$
12: compute target vector $\boldsymbol{V} = \frac{1}{K} \sum_k \boldsymbol{U}_k$
13: **for** $k = 1, 2, \ldots, K$ **do**
14:     compute RotoGrad's loss $L_k^{\text{roto}} = -\sum_n \langle \boldsymbol{R}_k^\top \widetilde{\boldsymbol{g}}_{n,k}, \boldsymbol{v}_n \rangle$
15:     update RotoGrad's parameters $\boldsymbol{R}_k = \boldsymbol{R}_k - \eta_{\text{roto}} \nabla_{\boldsymbol{R}_k} L_k^{\text{roto}}$
16:     update head's parameters $\boldsymbol{\phi}_k = \boldsymbol{\phi}_k - \eta \nabla_{\boldsymbol{\phi}_k} L_k$
17: **end for**

**Unconstrained optimization.** As previously discussed, parameters $\boldsymbol{R}_k$ are defined as rotation matrices, and thus the *Rotation* optimization in problem 4 is a constrained problem. While this would typically imply using expensive algorithms like Riemannian gradient descent [1], we can leverage recent work on manifold parametrization [5] and, instead, apply unconstrained optimization methods by automatically[2] parametrizing $\boldsymbol{R}_k$ via exponential maps on the Lie algebra of $SO(d)$.

**Memory efficiency and time complexity.** Second, as we need one rotation matrix per task, we have to store $O(Kd^2)$ additional parameters. In practice, we only need $Kd(d-1)/2$ parameters due to the aforementioned parametrization and, in most cases, this amounts to a small part of the total number of parameters. Moreover, as described by Casado et al. [5], parametrizing $\boldsymbol{R}_k$ enables efficient computations compared with traditional methods, with a time complexity of $O(d^3)$ independently of the batch size. In our case, the time complexity is of $O(Kd^3)$, which scales better with respect to the number of tasks than existing methods (for example, $O(K^2 d)$ for PCGrad [33]). Moreover, caching $\boldsymbol{R}_k$ in the forward pass and GPU parallelization can further reduce training time.

**Scaling-up RotoGrad.** Even though we can efficiently compute and optimize the rotation matrix $\boldsymbol{R}_k$, in some application domains, like computer vision, in which the size $d$ of the shared representation $\boldsymbol{z}$ is large, the time complexity for updating the rotation matrix may become comparable to the one of the network updates. In those cases, we propose to only rotate a subspace of the feature space, that is, rotate only $m << d$ dimensions of $\boldsymbol{z}$. Then, we can simply apply a transformation of the form $\boldsymbol{r}_k = [\boldsymbol{R}_k \boldsymbol{z}_{1:m}, \boldsymbol{z}_{m+1:d}]$, where $\boldsymbol{z}_{a:b}$ denotes the elements of $\boldsymbol{z}$ with indexes $a, a+1, \ldots, b$. While there exist other possible solutions, such as using block-diagonal rotation matrices $\boldsymbol{R}_k$, we defer them to future work.

# 4  Illustrative examples

In this section, we illustrate the behavior of RotoGrad in two synthetic scenarios, providing clean qualitative results about its effect on the optimization process. Appendix C.1 provides a detailed description of the experimental setups.

To this end, we propose two different multi-task regression problems of the form

$$L(\boldsymbol{x}) = L_1(\boldsymbol{x}) + L_2(\boldsymbol{x}) = \varphi(\boldsymbol{R}_1 f(\boldsymbol{x}; \boldsymbol{\theta}), 0) + \varphi(\boldsymbol{R}_2 f(\boldsymbol{x}; \boldsymbol{\theta}), 1), \tag{5}$$

where $\varphi$ is a test function with a single global optimum whose position is parametrized by the second argument, that is, both tasks are identical (and thus related) up to a translation. We use a single input

---

[2]For example, Geotorch [4] makes this transparent to the user.

$\boldsymbol{x} \in \mathbb{R}^2$ and drop task-specific network parameters. As backbone, we take a simple network of the form $\boldsymbol{z} = \boldsymbol{W}_2 \max(\boldsymbol{W}_1 \boldsymbol{x} + \boldsymbol{b}_1, 0) + \boldsymbol{b}_2$ with $\boldsymbol{b}_1 \in \mathbb{R}^{10}, \boldsymbol{b}_2 \in \mathbb{R}^2$, and $\boldsymbol{W}_1, \boldsymbol{W}_2^{\top} \in \mathbb{R}^{10 \times 2}$.

For the first experiment we choose a simple (avocado-shaped) convex objective function and, for the second one, we opt for a non-convex function with several local optima and a single global optimum. Figure 1 shows the training trajectories in the presence (and absence) of RotoGrad in both experiments, depicted as level plots in the space of $\boldsymbol{z}$ and $\boldsymbol{r}_k$, respectively. We can observe that in the first experiment (Figure 1a), RotoGrad finds both optima—which is in stark contrast to the vanilla case—by rotating the feature space and matching the (unique) local optima of the tasks. Similarly, the second experiment (Figure 1b) shows that, as we have two symmetric tasks and a non-equidistant starting point, in the vanilla case the optimization is dominated by the task with an optimum closest to the starting point. RotoGrad avoids this behavior by equalizing gradients and, by aligning gradients, is able to find the optima of both functions.

## 5   Related Work

Understanding and improving the interaction between tasks is one of the most fundamental problems of MTL, since any improvement in this regard would translate to all MTL systems. Consequently, several approaches to address this problem have been adopted in the literature. Among the different lines of work, the one most related to the present work is gradient homogenization.

**Gradient homogenization.** Since the problem is two-fold, there are two main lines of work. On the one hand, we have task-weighting approaches that focus on alleviating magnitude differences. Similar to us, GradNorm [6] attempts to learn all tasks at a similar rate, yet they propose to learn these weights as parameters. Instead, we provide a closed-form solution in Equation 1, and so does IMTL-G [18]. However, IMTL-G scales all task gradients such that all projections of $\boldsymbol{G}$ onto $\boldsymbol{G}_k$ are equal. MGDA [28], instead, adopts an iterative method based on the Frank-Wolfe algorithm in order to find the set of weights $\{\omega_k\}$ (with $\sum_k \omega_k = 1$) such that $\sum_k \omega_k \boldsymbol{G}_k$ has minimum norm. On the other hand, recent works have started to put attention on the conflicting direction problem. Maninis et al. [22] first proposed adversarial training to make task gradients statistically indistinguishable as part of a bigger image-tailored architecture. More recently, PCGrad [33] proposed to drop the projection of one task gradient onto another if they are in conflict, whereas GradDrop [7] randomly drops elements of the task gradients based on a sign-purity score.

In the literature, we can also find other approaches which, while orthogonal to the gradient homogenization, are **complementary to our work** and thus could be used along with RotoGrad. Next, we provide a brief overview of them.

A prominent approach for MTL is task clustering, that is, selecting which tasks should be learned together. This approach dates back to the original task-clustering algorithm [31], but new work in this direction keeps coming out [29, 35]. Alternative approaches, for example, scale the loss of each task differently based on different criteria such as task uncertainty [14], task prioritization [11], or similar loss magnitudes [18]. Moreover, while most models fall into the hard-parameter sharing umbrella, there exists other architectures in the literature. Soft-parameter sharing architectures [27], for example, do not have shared parameters but instead impose some kind of shared restrictions to the entire set of parameters. An interesting approach consists in letting the model itself learn which parts of the architecture should be used for each of the tasks [12, 23, 30, 32]. Other architectures, such as MTAN [19], make use of task-specific attention to select relevant features for each task. Finally, problems triggered by the differences between task gradients (in magnitude and direction) have also been studied in other domains like meta-learning [34] and continual learning [21].

## 6   Experiments

In this section we assess the performance of RotoGrad on a wide range of datasets and MTL architectures. First, we check the effect of the learning rates of the rotation and network updates on the stability of the learning process of RotoGrad. Then, with the goal of applying RotoGrad in scenarios with extremely large sizes of $\boldsymbol{z}$, we explore the effect of rotating a subspace of $\boldsymbol{z}$ instead of the whole shared representation. Finally, we compare our approach with competing MTL solutions in the literature, showing that RotoGrad consistently outperforms all other methods. Refer to Appendix C for a more detailed description of the experimental setups and additional results.

242 **Relative task improvement.** Since MTL uses different metrics for different tasks, throughout this
243 section we group results by means of the relative task improvement, first introduced in [22]. Given a
244 task $k$, and the metrics obtained during test time by our model, $M_k$, and by a baseline model, $S_k$,
245 which consists of $K$ networks trained on each task individually, the relative task improvement for the
246 $k$-th task is defined as

$$\Delta_k \coloneqq 100 \cdot (-1)^{l_k} \frac{M_k - S_k}{S_k}, \tag{6}$$

247 where $l_k = 1$ if $M_k < S_k$ means that our model performs better than the baseline in the $k$-th task, and
248 $l_k = 0$ otherwise. We depict our results using different statistics of $\Delta_k$ such as its mean ($\text{avg}_k \Delta_k$),
249 maximum ($\max_k \Delta_k$), and median ($\text{med}_k \Delta_k$) across tasks.

## 6.1 Training stability

251 At the end of Section 3.2 we discussed that, by casting
252 problem 4 as a Stackelberg game, we have convergence
253 guarantees when the rotation optimizer is the slow-learner.
254 Next, we empirically show this necessary condition.

255 **Experimental setup.** Similar to [28], we use a multi-task
256 version of MNIST [16] where each image is composed
257 of a left and right digit, and use as backbone a reduced
258 version of LeNet [17] with light-weight heads. Besides
259 the left- and right-digit classification proposed in [28], we
260 consider three other quantities to predict: i) sum of digits;
261 ii) parity of the digit product; and iii) number of active
262 pixels. The idea here is to enforce all digit-related tasks

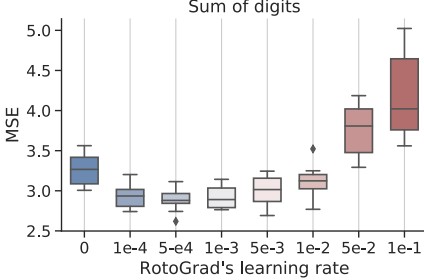

Figure 3: Test error on the sum of digits task for different values of RotoGrad's learning rate.

263 to cooperate (positive transfer), while the (orthogonal) image-related task should not disrupt these
264 learning dynamics. We use negative cross-entropy and accuracy for the left- and right-digit tasks,
265 binary cross-entropy and f1-score for the parity task, and mean squared error (MSE) as loss and
266 metric for both regression tasks.

267 **Results.** Figure 3 shows the effect averaged over ten independent runs—in terms of test error in the
268 sum task, while the rest of tasks are shown in Appendix C.2—of changing the rotations' learning rate.
269 We can observe that, the bigger the learning rate is in comparison to that of the network's parameters
270 (1e−3), the higher and more noisy the test error becomes. MSE keeps decreasing as we lower the
271 learning rate, reaching a sweet-spot at half the network's learning rate (5e−4). For smaller values,
272 the rotations' learning is too slow and results start to resemble those of the vanilla case, in which no
273 rotations are applied (leftmost box in Figure 3).

## 6.2 Rotating a subspace

275 Next, we evaluate the effect of subspace rotations as described at the end of Section 3.4, assessing the
276 trade-off between avoiding negative transfer and size of the subspace considered by RotoGrad.

277 **Experimental setup.** We test RotoGrad on a 10-task classification problem on CIFAR10 [15], using
278 binary cross-entropy and f1-score as loss and metric, respectively, for all tasks. We use ResNet18 [13]
279 without pre-training as backbone ($d = 512$), and linear layers with sigmoids as task-specific heads.

280 **Results** are summarized at the bottom part of Table 1. We can observe that rotating the entire space
281 provides the best results, and they worsen as we decrease the size of $\boldsymbol{R}_k$. However, rotating only 64
282 features (12.5 % of the shared feature space) still yields better results than vanilla optimization.

## 6.3 Methods comparison

284 We now proceed to compare RotoGrad with the different existing approaches to gradient conflict (for
285 both magnitude and direction) in different real-world datasets, showing how RotoGrad outperforms
286 existing methods while being on par with existing methods in training time.

287 **Experimental setup.** In order to provide fair comparisons among methods, all experiments use
288 identical configurations and random initializations. For all methods we performed a hyper-parameter
289 search and chose the best ones based on validation error. Our results are reported using the median and
290 standard deviation computed over 5-10 random seeds. Further details can be found in Appendix C.1.

Table 1: Task performance on CIFAR10 for different competing methods (top) and RotoGrad with matrices $R_k$ of different sizes (bottom). Table shows median and standard deviation over five runs.

| Method | $\text{avg}_k \Delta_k \uparrow$ | $\text{med}_k \Delta_k \uparrow$ | $\text{max}_k \Delta_k \uparrow$ |
|---|---|---|---|
| Vanilla | $2.58 \pm 0.54$ | $2.73 \pm 1.37$ | $11.14 \pm 3.35$ |
| GradDrop | $3.07 \pm 0.48$ | $3.18 \pm 1.07$ | $14.03 \pm 2.83$ |
| PCGrad | $2.86 \pm 0.81$ | $3.33 \pm 1.68$ | $12.01 \pm 3.19$ |
| MGDA | $-1.75 \pm 0.43$ | $-4.48 \pm 2.35$ | $3.67 \pm 0.98$ |
| GradNorm | $-0.08 \pm 0.95$ | $0.09 \pm 2.23$ | $8.82 \pm 3.41$ |
| IMTL-G | $2.73 \pm 0.27$ | $1.95 \pm 2.21$ | $10.20 \pm 2.98$ |
| IMTL-G+$R_k$ | $3.02 \pm 0.69$ | $4.38 \pm 1.11$ | $12.76 \pm 1.77$ |
| RotoGrad 64 | $2.90 \pm 0.49$ | $3.44 \pm 1.51$ | $13.16 \pm 2.40$ |
| RotoGrad 128 | $2.97 \pm 1.08$ | $3.73 \pm 2.14$ | $12.64 \pm 3.56$ |
| RotoGrad 256 | $3.68 \pm 0.68$ | $3.29 \pm 2.18$ | $14.01 \pm 3.22$ |
| RotoGrad 512 | $\mathbf{4.48 \pm 0.99}$ | $\mathbf{4.72 \pm 2.84}$ | $\mathbf{15.57 \pm 3.99}$ |

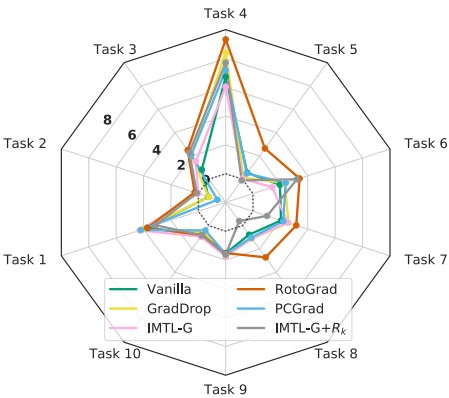

Figure 4: Task improvement (median over five runs) of different methods on CI-FAR10. RotoGrad outperforms competing methods on all tasks.

**MNIST and SVHN.** We reuse the experimental setting from Section 6.1—now with multi-task versions of MNIST [16] and SVHN [26]—in order to evaluate how disruptive the orthogonal image-related task is for different methods. We can observe in the results from Table 2 that the effect of the image-related task is more disruptive in MNIST, in which MGDA utterly fails. Direction-aware methods (Grad-Drop and PCGrad) do not improve the vanilla results, whereas IMTL-G, GradNorm, and RotoGrad obtain the best results.

Table 2: Test performance (median and standard deviation) on two set of unrelated tasks, across ten different runs.

| | MNIST | | SVHN | |
| | Digits | Act Pix | Digits | Act Pix |
| Method | $\text{avg}_k \Delta_k \uparrow$ | MSE $\downarrow$ | $\text{avg}_k \Delta_k \uparrow$ | MSE $\downarrow$ |
|---|---|---|---|---|
| Single | - | $0.01 \pm 0.01$ | - | $0.17 \pm 0.06$ |
| Vanilla | $-2.51 \pm 3.01$ | $0.11 \pm 0.01$ | $5.14 \pm 0.83$ | $2.75 \pm 3.17$ |
| GradDrop | $-2.51 \pm 1.73$ | $0.13 \pm 0.02$ | $5.68 \pm 1.05$ | $1.91 \pm 0.86$ |
| PCGrad | $-3.12 \pm 3.88$ | $0.12 \pm 0.02$ | $5.50 \pm 0.75$ | $2.26 \pm 0.85$ |
| MGDA | $-12.57 \pm 9.97$ | $0.06 \pm 0.02$ | $5.99 \pm 1.48$ | $0.66 \pm 0.75$ |
| GradNorm | $0.13 \pm 2.27$ | $0.08 \pm 0.01$ | $6.67 \pm 1.02$ | $1.41 \pm 0.74$ |
| IMTL-G | $1.17 \pm 2.77$ | $0.07 \pm 0.01$ | $5.81 \pm 0.85$ | $2.47 \pm 1.65$ |
| RotoGrad | $2.12 \pm 2.23$ | $0.08 \pm 0.02$ | $6.08 \pm 0.48$ | $1.61 \pm 2.72$ |

**CIFAR10.** We reuse the setting in Section 6.2 and compare the different MTL methods using five different seeds. Results are shown in Table 1 and Figure 4. Unlike the previous setting, scaling gradients is not enough to solve the problem. Among existing methods, both direction-aware solutions (PCGrad and GradDrop) improve over the vanilla case on all the statistics, whereas most magnitude-aware solutions substantially worsen task performance. In stark contrast, RotoGrad improves task performance across all ten tasks, as it can be observed both in Table 1 and Figure 4. To further show that this is a consequence of gradient homogenization in terms of both magnitudes and directions, we introduced an extra-baseline, IMTL-G+$R_k$, which applies IMTL-G to the extended MTL architecture (Figure 2), that is, with matrix $R_k$ optimizing the $k$-th task loss (instead of Equation 3).

**NYUv2.** Now, we test all methods using NYUv2 [9] on three different tasks: 13-class semantic segmentation; depth estimation; and surface normals. To speed up training, all images were resized to $288 \times 384$ resolution; and data augmentation was applied to alleviate overfitting. As MTL architecture, we use SegNet [2] where the decoder is splitted into three convolutional heads. We use the same setup as Liu et al. [19]. Like in previous experiments, we observe in Table 3 that RotoGrad results in a consistent improvement over all tasks with respect to the vanilla case. MGDA obtains the best results in surface normals at the expense of overlooking the other tasks, while GradDrop worsens all results and PCGrad obtains minor improvements in all tasks. GradNorm finds a trade-off solution instead, improving results in depth estimation and surface normals, yet with worse results in semantic segmentation. RotoGrad obtains the best results followed by IMTL-G and, more importantly, RotoGrad is the only method resulting in a average positive task improvement—across the three tasks—over training three single-task models independently. It is worth mentioning that, with only

Table 3: Results for different methods on the NYUv2 dataset with a SegNet model. RotoGrad obtains the best performance in segmentation and depth tasks on all metrics, while significantly improving the results on normal surfaces with respect to the vanilla case.

| Method | Semantic Segmentation ↑ | | | Depth Estimation ↓ | | | Surface Normal | | | | | | Hours |
|---|---|---|---|---|---|---|---|---|---|---|---|---|---|
| | | | | | | | Angle Distance ↓ | | Within $t°$ ↑ | | | | |
| | mIoU | Pix Acc | $\text{avg}_k \Delta_k$ ↑ | Abs Err | Rel Err | $\text{avg}_k \Delta_k$ ↑ | Mean | Median | 11.25 | 22.5 | 30 | $\text{avg}_k \Delta_k$ ↑ | |
| Single | 0.38 | 0.63 | - | 0.59 | 0.23 | - | 24.76 | 18.99 | 30.11 | 57.81 | 69.90 | - | 11.37 |
| Vanilla | 0.37 | 0.64 | −0.62 | 0.56 | 0.22 | 3.68 | 30.09 | 26.09 | 19.74 | 43.62 | 57.07 | −27.26 | **3.45** |
| GradDrop | 0.37 | 0.63 | −1.55 | 0.59 | 0.24 | −2.22 | 30.81 | 27.19 | 17.68 | 41.44 | 55.15 | −31.67 | 3.55 |
| PCGrad | 0.39 | 0.64 | 1.50 | 0.54 | 0.22 | 4.99 | 29.85 | 25.81 | 19.41 | 44.02 | 57.64 | −26.68 | 3.51 |
| MGDA | 0.20 | 0.51 | −32.75 | 0.73 | 0.28 | −22.33 | **24.98** | **19.02** | **30.57** | **57.61** | **69.41** | **−0.11** | 3.55 |
| GradNorm | 0.36 | 0.64 | −1.74 | 0.55 | 0.23 | 3.31 | 25.80 | 20.30 | 28.22 | 54.91 | 67.21 | −5.25 | 3.54 |
| IMTL-G | 0.38 | 0.65 | 1.92 | 0.55 | 0.23 | 3.64 | 26.83 | 21.96 | 25.14 | 51.74 | 64.76 | −11.67 | 3.60 |
| RotoGrad | **0.40** | **0.66** | **5.33** | **0.54** | **0.20** | **9.06** | 26.35 | 21.27 | 26.25 | 53.11 | 65.99 | −8.99 | 3.85 |

Table 4: Task f1-score statistics and training hours in CelebA for all competing methods and two different architectures/settings. RotoGrad obtains the best performance in both setups with comparable training time as existing methods.

| Method | Convolutional ($d = 512$) task f1-scores (%) ↑ | | | | | ResNet18 ($d = 2048$) task f1-scores (%) ↑ | | | | |
|---|---|---|---|---|---|---|---|---|---|---|
| | $\text{min}_k$ | $\text{med}_k$ | $\text{avg}_k$ | $\text{std}_k$ ↓ | Hours | $\text{min}_k$ | $\text{med}_k$ | $\text{avg}_k$ | $\text{std}_k$ ↓ | Hours |
| Vanilla | 1.62 | 54.74 | 58.69 | 24.18 | 4.06 | 15.45 | 61.52 | 61.25 | 22.09 | 1.49 |
| GradDrop | 3.94 | 55.80 | 58.62 | 23.98 | 4.42 | 4.46 | 63.52 | 63.61 | 21.79 | 1.60 |
| PCGrad | 2.69 | 60.30 | 59.83 | 23.85 | 17.03 | 17.23 | 61.82 | 62.74 | 20.84 | 5.90 |
| GradNorm | 1.83 | 52.17 | 54.68 | 24.94 | 11.02 | 14.43 | **64.10** | 63.51 | 21.20 | 3.59 |
| IMTL-G | 3.31 | 53.05 | 56.05 | 26.92 | 4.90 | 21.52 | 62.12 | 61.98 | 21.62 | 1.72 |
| RotoGrad | **9.11** | **62.31** | **62.45** | **22.14** | 11.00 | **25.72** | 63.84 | **65.17** | **18.99** | 6.90 |

three tasks, all methods trained in less than 4 hours; and that this result consolidates RotoGrad's scalability, as we only rotate the first $1024$ dimensions of $\boldsymbol{z}$, out of a total of 7 millions.

**CelebA.** Last, we apply all methods to a 40-class multi-classification problem in CelebA [20] on two different settings: one using a convolutional network as backbone ($d = 512$); and another using ResNet18 [13] as backbone ($d = 2048$). Similar to CIFAR10, we use binary cross-entropy and f1-score as loss and metric for all tasks. Even though we face two completely different architectures, results in Table 4 show that RotoGrad convincingly outperforms all competing methods in all f1-score statistics, independently of the model. Furthermore, since this is a computationally demanding task with 40 tasks—in fact, we omit MGDA as it takes several days to train—we also compare methods in terms of training time. On the one hand, GradDrop and IMTL-G produce little overhead compared with the vanilla case, as expected. On the other hand, GradNorm and PCGrad take, respectively, 2.5 and 4 times longer to train than the vanilla setting. More importantly, RotoGrad outperforms existing methods while staying on par with them in training time, rotating $50\,\%$ and $75\,\%$ of the shared feature $\boldsymbol{z}$ for the convolutional and residual backbones, respectively, which further demonstrates that RotoGrad can scale-up to real-world settings.

# 7 Conclusions

In this work, we have introduced RotoGrad, an algorithm that tackles negative transfer in MTL by homogenizing task gradients in terms of both magnitudes and directions. RotoGrad enforces a similar convergence rate for all tasks, while at the same time smoothly rotates the shared representation differently for each task in order to avoid conflicting gradients. As a result, RotoGrad leads to stable and accurate MTL. Our empirical results have shown the effectiveness of RotoGrad in many scenarios, staying on top of all competing methods in performance, while being on par in terms of computational complexity with those that better scale to complex networks.

We believe our work opens up interesting venues for future work. For example, it would be interesting to study alternative approaches to further scale up RotoGrad using, for example, diagonal-block or sparse rotation matrices; to rotate the feature space in application domains with structured features (e.g., channel-wise rotations in images); and to combine different methods, for example, by scaling gradients using the direction-awareness of IMTL-G and the "favor slow-learners" policy of RotoGrad.

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
