# OpenReview forum: "RotoGrad: Gradient Homogenization in Multi-Task Learning"
_NeurIPS.cc/2021/Conference — NeurIPS 2021 Submitted_

### Official Review · Reviewer_Kz4R · 2021-07-07

**Rating:** 4
**Confidence:** 5

**Summary:**

The work proposes RotoGrad, a multitask layer that rotates a shared feature space in a deep neural network to align gradients and mitigate negative transfer effects. The proposed method works within models with multiple prediction heads and a shared backbone, and deals both with magnitude and directional imbalance between task gradients. Authors provide multiple experiments in the computer vision space to show efficacy of the proposed method against other multitask baselines.


**Limitations And Societal Impact:**

More discussion could be provided regarding limitations and societal impact. In terms of the former, for example, there are plenty of benchmarks on which RotoGrad seems to not perform the best, so discussion of why needs to be more in detail.

**Main Review:**

**Overall:** The paper presents an interesting new method to treat multitask learning systems. If robust, this method can have general use to the community and be a powerful addition to the toolbox. However, there are many unanswered questions and I remained overall unconvinced that the method works without major additional experimentation, including detailed ablation studies, and justifications (too many to address within the short rebuttal period in my opinion). I do think the method shows promise though, so regardless of the decision after this review process I hope the authors continue to work on their method.

**Pros:**

(P1) Method is clear and intuitive, and tackles a crucial problem within optimal multitask training.

(P2) Results seem promising and beat other baselines on benchmarks within the main paper.

(P3) Formulation of the method as a Stackelberg game is interesting and can perhaps provide general insight into these problems, as many modern MTL methods involve optimizing auxiliary losses.

(P4) Results are given for multiple model structures and multiple datasets performing the same tasks.

(P5) Code is provided for reproducibility.

**Needs Comment/Revision (major):**

(RM1) Why must the proposed method interact with the forward pass? There are some other methods (see for example Sinha et al. at https://arxiv.org/abs/1806.08028) that propose methods which only interact with the backwards pass and seem to also work well. Either theoretical justification or additional experiments are needed to justify why doing the aggressive thing of altering the forward pass feature space is necessary.

(RM2) The toy example provided is unconvincing, as the only comparison is to vanilla SGD. It is rather puzzling that even the Rotograd layer within the toy example is optimized using RAdam, which I feel would give Rotograd a severe advantage.

(RM3) When the proposed method outperforms other methods that’s great, but when all the other methods perform very poorly it raises questions of whether the other methods are implemented correctly. For instance, why is MGDA performing so poorly on the CIFAR/MGDA tasks when the original MGDA paper used that as a benchmark and actually performed quite well? These discrepancies with the prior results need to be addressed for the current results to be fully believed.

(RM4) One strange detail is that GradNorm and GradDrop seem to have different compute overheads within the current implementations, at least within the CelebA experiments. However, as far as I can tell, GradNorm and GradDrop have essentially the same compute overhead (with the limiting factor in the compute overhead the same computation of dL_i/dw for each task loss L_i), so this discrepancy makes me concerned that the present implementations of baselines is not correct. Similarly, RotoGrad seems to be even slower than PCGrad, and PCGrad has to calculate dL_i/dw for all trainable weights while RotoGrad only needs to backpropagate to the feature layer, so this overhead is somewhat concerning as well.

(RM5) Ablation studies are needed to see whether both the magnitude and directional components of Rotograd are both needed.

(RM6) The magnitude part of the implementation seems to include the initial gradient magnitude as part of the normalized quantity. However, this is incredibly dependent on the network initialization. Did authors account for this in some way?

(RM7) The digit classification task performances seem to be very dependent on the size of the RotoGrad layer. Looking at the appendix, the network structure for these tasks is very shallow, so how can the authors rule out the effect of the added network capacity within the RotoGrad layer?

**Needs Comment/Revision (less major):**

(RLM1) Proposition 3.1 is correct but is rather obvious - would authors consider rephrasing into nonmathematical language to reduce potential confusion? If two gradients have positive pairwise similarity, then following their sum (for small enough learning rate) will clearly reduce both losses because loss reduction is step_size*(grad_k dot grad_sum). This proposition has also been heavily treated in prior work as well like MGDA and PCGrad, so does not necessarily need independent treatment here and can just be stated.

(RLM2) Authors claim that homogenizing gradients at one layer is equivalent to homogenizing gradients at the weight layers. This is a false statement, as chained gradient computations will cause even initially homogenized gradients at a feature layer to increasingly stray from each other. Authors must justify why they can ignore this effect.

(RLM3) Figures need to be placed closer to the text where they’re referenced.

(RLM4) Sometimes bolded numbers in tables are not the best results for that column.

*Post rebuttal*: I thank the authors for their time and effort in crafting their response. A lot of my more technical comments were addressed. I am still very concerned about the extra capacity and don't feel these concerns are properly addressed by pointing out that one other baseline (IMTL-G) also has added capacity, and some of the new results actually may run counter to the proposed narrative - for example RotoGradNorm beating out RotoGrad. The method itself is well defined but I am still not sold on the scientific validity of the analysis. I will keep my rating a 4.


**Time Spent Reviewing:**

2

---

> ### Author Response · Authors · 2021-08-10
> **Response to reviewer Kz4R**
>
> We thank reviewer Kz4R for the valuable feedback, which has helped us to further improve the paper. Before entering with the individual comments, we would like to remark that empirically validating RotoGrad has been (and is still) a main objective for us, and that is why we have invested significant time and computational resources to include 4 different  architectures and 5 datasets, several independent runs with different initializations when possible, and now also the requested ablation study. We believe that these experiments support our main contributions, and we fail to completely understand what the reviewer means with "major additional experimentation".
>
> **(RM1) Why must the proposed method interact with the forward pass?** First, we thank reviewer Kz4R for sharing this prior work, as we missed it in our literature review; we will include it in the discussion as a closely related work.
>
> Interacting in the forward pass has several benefits compared with working on the backward pass. First, it offers fine control on the type of transformation applied to the gradients (in this case, rotations, which preserve distances). Second, we believe it is a simple and natural choice (“if we want to change a gradient direction, we can simply modify its coordinate system”) which, in turn, _passively_ modifies the per-task gradients (to avoid conflicts) without the need of changing the backward pass. Finally, RotoGrad’s overhead comes from the extra parameters, but it avoids having to backpropagate a second time through the network’s computational graph, which is the main overhead of backward-only methods (including the one shared by the reviewer).  We  believe that this precisely is the main source of  novelty and also improved performance of RotoGrad. We will further discuss this point in the revised version of the paper to better motivate RotoGrad.
>
> **(RM2) Toy example. RAdam in RotoGrad.** We would like to remark that the main purpose of these toy examples is to intuitively illustrate RotoGrad, i.e.,  to help the reader understand how RotoGrad works, in comparison with vanilla training. However, we will gladly add to the appendix the same toy examples using the baselines.
>
> With respect to the experimental set up, let us clarify that the network’s parameters are optimized using SGD in both cases, while RAdam is only used to update the rotation parameters. In all our experiments, we use RAdam (instead of RMSProp, which was originally proposed to optimize rotations in NNs [5]) for updating the rotation parameters because it shows a more robust behavior than SGD. Yet, we are happy to include in the appendix experiments with other optimizers than SGD (e.g., RAdam, ADAM, ...) for the network parameters updates in the toy experiment. For real-world datasets, as specified in the Appendix, we have used the same optimizers for all baselines.
>
> **(RM3) Baselines’ implementations.** To the best of our knowledge, all the baselines that we compare  here, but MGDA, do not provide an official code implementation. We have carefully followed their method descriptions, extensively tested each of them during experimentation, ran several seeds on most experiments, provided several performance metrics (e.g. f1 instead of accuracy for imbalance datasets such as CelebA), as well as provided the code. As stated in the paper, we will publicly release our code for the sake of transparency and reproducibility.
>
> Regarding the reviewer's example on MGDA, here we use the official implementation of  MGDA, which we took and included in our codebase. Although we could not find any CIFAR experiments on the original  MGDA paper [28], we want to remark that our results on MGDA are on par with those shown in recent literature, e.g., IMTL-G [18]. The authors of [18] analyse MGDA and conclude that `MGDA focuses on tasks with smaller gradient magnitudes`. In other words, MGDA performs well in some tasks and overlooks others. The consistency of our results can be easily verified by comparing Tables 1 and 3 from our paper with Tables 1 and 2 from IMTL-G [18].
>
> **(RM4.1) Overhead comparison between GradNorm and GradDrop.** To the best of our understanding, these two methods do not have the same computational overhead. This can be explained mainly due to the fact that GradNorm has learnable parameters per task for which we need to compute the loss and backpropagate through. This can be seen in our (37-line) implementation of GradNorm (methods/gradnorm.py). As a result, GradNorm needs,  in the case of e.g. CelebA, to backpropagate once per each of the 40 tasks. In contrast, GradDrop is a light-weight method with no learnable parameters (refer to methods/graddrop.py), which was indeed one of its selling points in the original paper (Table 1 in [7]).
>
> **(RM4.2) Overhead in PCGrad.** In order to provide a fair comparison with the rest of methods, we adopted a similar approach as other recent works [7] and implemented PCGrad on the last shared representation $z$ (instead of on the network parameters). Unfortunately,  _we cut the explicit mention to this point during the final editing of the paper by mistake, and it was only mentioned in an implicit way_ (see that, e.g., we talked of PCGrad in terms of d in L173). We will make sure to bring this detail back in the next revision.
>
> **(RM5) Ablation study.** Please refer to the [general reply](https://openreview.net/forum?id=J_VRu0ZFVX-&noteId=RK8bbD33dpb).
>
> **(RM6) Dividing by the initial gradient magnitude.** We agree with the reviewer that indeed the initial gradient magnitude heavily depends on the network initialization. To account for this, we use a warm-up period of 20 iterations in our implementation. After 20 iterations, we reassign the value of $G_k^0$ in Eq. (2), thus dividing by the gradient in the 20-th iteration, $G_k^{20}$. While simple, this approach works well in practice, and it is similar to approaches observed in implementations of loss normalization in MTL code.
>
> **(RM7) Extra capacity.** We are not sure that we fully understand this point, but let us give it a try. As we agree that concerns regarding the effect of the extra parameters are sensible, for that very same reason we introduced IMTL-G+$​R_k$ on the CIFAR10 experiments (where all are classification tasks). This baseline (explained in L314-315) uses the latest MTL baseline, IMTL-G, with a network that includes rotation matrices as free parameters (each optimizing their task-loss instead of Eq. 3). As Table 1 shows, RotoGrad with the same number of parameters clearly outperforms this baseline in all statistics. Even more, it shows that IMTL-G+$R_k​$ deteriorates tasks performance with respect to the IMTL-G (see Task 7-8 in Figure 4), despite the extra capacity. This allows us to conclude that the improvement of RotoGrad does not come (only) from the extra parameters but from the Rotograd optimization (i.e., Eq. 3).
>
> **(RLM1) Prop. 3.1.** We added this proposition to describe the interplay between both proposed algorithms to the reader. We agree that it is a simple statement that could alternatively be explained more intuitively in plain English. Thus, we will happily move the proposition to the Appendix in favor of a more readable discussion in the revised version of the paper.
>
> **(RLM2) Justify homogenizing in the last shared feature.** We agree with the reviewer than homogenizing gradients at the shared feature $z$ is only equivalent to homogenizing the gradients with respect to the network parameters, when they are completely homogenized (such that $​g_k = g$ for all $k$), which is what we actually meant in L77 (we will clarify this point). Yet, we remark that optimizing w.r.t $​z$ is a common practice (see IMTL-G [19], GradDrop [7], GradNorm [6]), which is justified as a way of reducing the computational cost. We will add these arguments to the discussion in L77.
>
> **(RLM3) Figures need to be placed closer to the text where they are referenced.** We believe the reviewer refers to Figure 1. As mentioned before, we designed toy examples to help the reader easily understand how RotoGrad works. For that reason, we deliberately put said figure early in the paper, so that the reader had an opportunity to gain an intuition on RotoGrad before getting into details. However, if the reviewer thinks that moving the figure closer to its description would improve readability, we will happily do so in the revised paper
>
> **(RLM4) Sometimes bolded numbers in tables are not the best results for that column.** We believe the reviewer refers to Table 3. We will revisit all the tables to make sure that our highlighting of the results is consistent across methods and compatible with NeurIPS guidelines.

---

### Official Review · Reviewer_3UmL · 2021-07-11

**Rating:** 7
**Confidence:** 4

**Summary:**

This paper proposes RotoGrad, a multi-task optimisation strategy that normalises the gradient magnitude and (conflicting) direction in a unified manner, to avoid the negative transfer. The proposed method is intuitive, theoretically motivated by the two player-game, and hyper-parameter free (except for learning rate required for updating rotation parameters).

The paper extensively evaluates the proposed method comparing with other baselines, both in single domain MTL setting NYUv2, and in multi-domain MTL setting CIFAR-10, and CelebA, and have showed improved performances in all settings.

**Limitations And Societal Impact:**

This paper studies the theoretical insights and optimisation strategy for multi-task learning, and thus having no societal impact.

**Main Review:**

The paper is clearly written, and well-motivated from the existing Stackelberg leadership model. I also appreciate the authors discuss the memory and time complexity in Section 3.4, and being honest about the current limitation on scaling issues in computer vision tasks, which typically with large representation dimensions, requiring extensive computing time for an exact solution without any approximation.

Generally, I like the idea and particularly for the part that introducing the gradient rotation, which I believe is novel and overlooked in previous multi-task optimization papers I know. Here are some other comments, which I hope the authors could clarify and hopefully could further improve the paper.

-- Subspace rotation. As mentioned in the previous comment, for computing a direct solution requires a $d^2$ rotation matrix for each task with representation dimension $d$. The introduced rotation parameters can be neglectable if $d$ is small. However, if $d$ is large as typically in most computer vision tasks, it introduces more parameters compared to baseline methods, which leads to unfair comparison.  We thus would not know whether the improved performance is based on the design of this method, or simply because of the increased number of learnable parameters. In Table 1, applying the rotation only in the 64 subspace is having a close and worse performance compared to the baseline performance, this verifies my concern here. To be clear on the parameter size and/or FLOPs in each method would be essential for a fair comparison. Maybe adding a linear embedding to map to a smaller embedding size, or applying a block-wise rotation would alleviate this issue?

-- Limitation on multi-task network design. From my understanding, the proposed RotoGrad requires a clear top-down multi-task network design with a **single** shared representation output $z$, which the task-specific rotation function needs to be applied on. It would be tricky, or impossible? to apply this strategy to any design of the network, say MTAN as discussed in the related work, or other popular multi-task network design: such as AdaShare, Cross-Stitch network, from which the shared network parameters are defined implicitly. In this way, the RotoGrad would be very hard to optimise, maybe requiring network-specific modification on choosing the appropriate $z$. This is not a serious issue though, if the improved performance solely on hard parameter sharing is significant enough that surpass the improvements brought by the design of the network itself. A paragraph discussing this issue would be helpful.




**Time Spent Reviewing:**

3

---

> ### Author Response · Authors · 2021-08-10
> **Response to reviewer 3UmL**
>
> We appreciate the positive review and the reviewer's acknowledgement of our efforts to be as transparent  as possible when it comes to the assumptions and limitations of our work.
>
> **Extra capacity.** It is sensible to be concerned regarding the effect of the extra parameters and how fair comparisons are. We had this same concern, and that is why we decided to include IMTL-G+$R_k$ as a baseline in CIFAR10 (section 6.3). This extra baseline uses the latest method (IMTL-G) on an extended model with _full-size_ task-specific rotation matrices that are freely optimized (i.e. they minimize their task loss). IMTL-G+$R_k$ has the same number of parameters as RotoGrad 512, yet it performs significantly worse (see Table 1). This is even clearer when comparing IMTL-G, IMTL-G+$R_k$, and RotoGrad in Figure 4. These results allow us to conclude that the improvement of RotoGrad does not come (only) from the extra parameters but from the Rotograd optimization (i.e., Eq. 3). We will add further information on the  number of parameters of each method in the revised paper.
>
> **Subspace rotations.** We are afraid that we fail to see the reviewer's concerns. As discussed in section 6.2, we find the results shown in Table 1 encouraging, rather than concerning. At first, RotoGrad is meant to be used on the entire feature-space of $z$. However, results in Table 1 show that rotating only 12.5% of the space already yields better results that _all existing baselines_ (note that IMTL-G+$R_k$ is introduced here and uses full-size rotations), and results keep improving as we increase the rotation size. This allows practitioners to exploit the improved performance of RotoGrad while keeping the computational overhead under control.
>
> **Other architectures.** To the best of our knowledge, this is not an exclusive problem of RotoGrad. Combining algorithmic solutions (RotoGrad, GradNorm, MGDA, etc) with other architectures different from hard-parameter sharing is an under-explored area. While hard-parameter sharing architectures offer a straightforward use-case, as long as there are  shared features (such as $z$ in this work), it should be possible to apply RotoGrad in other MTL architectures (e.g. in MTAN). In the case of Adashare, one could apply RotoGrad at the end of shared blocks, but assumptions such as the ones in L75-78 would not hold any more. While we defer experiments in this direction  for future work, we will add a paragraph discussing how to possibly combine these two types of solutions in the revised paper.

---

> > ### Comment · Reviewer_3UmL · 2021-08-23
> > **Reply to Authors**
> >
> > I'd like to thank the authors' comments and clarification. And I decide to hold my current rating as my final decision.
> >
> > Only one small issue though: "Combining algorithmic solutions (RotoGrad, GradNorm, MGDA, etc) with other architectures different from hard-parameter sharing is an under-explored area." This is not strictly speaking correct: Any weighting-based MTL optimisation methods such as: Uncertainty [14] and DWA [in MTAN] are network agnostic. The MGDA, the multi-objective-based optimisation method is in theory also network agnostic, (just they did not evaluate different architectures in the original paper).

---

### Official Review · Reviewer_ab65 · 2021-07-14

**Rating:** 6
**Confidence:** 3

**Summary:**

The paper proposes a novel method for the multitask learning environment, which deals with the conflicting gradients problem (AKA negative transfer). The method comprises two parts for dealing with gradient magnitudes and directions. The latter part is quite novel, well-motivated (with a cartoon example from Fig. 1), and theoretically sound. The method is tested against a number of MTL benchmarks and is shown to produce better-performing models. Overall, the paper presents interest to the MTL community.

**Main Review:**

L27: why is gradient cancellation seen as a problem? If a model cannot explain training data, conflicting gradients can be seen as reducing the variance of the solution, yielding a model which is "equally" bad wrt multiple objectives (as shown in Fig.1). In such a case, the proposed solution can be seen as an efficient way to increase model capacity in the right places (where the heads branch out), thus leading to a model with higher capacity (which is not bad per se but requires acknowledgment).

L35: can prioritization of the "lagging" tasks be harmful, e.g., in cases where the task loss is noisy (e.g., due to extreme label noise)?

The usage of a task-specific rotation matrix is interesting; however, could it be replaced with a simpler learned construct, such as a task-conditional batch norm, keeping the alternating training scheme from Alg.1 intact?

The usage of matrix exponential for parameterizing the rotation matrix is interesting; however, in the light of the argument in L151, do the gradients of the unconstrained parameters of R require any treatment (gradient magnitude/LR adjustment)?

One unclear aspect associated with matrices R is whether the preceding activations are the output of a linear layer or a nonlinearity. If activations are nonlinear and given that such a matrix R constitutes an additional linear layer, it would be interesting to see extended single-task baselines with matrix R added in the right place (where it would otherwise branch out from the backbone in MTL).

An ablation study of the magnitude and direction counterparts, particularly where another more traditional method can be applied in place of one of the proposed magnitude components, is missing. The closest I could find in Table 1 reporting IMTL-G+$R_k$

================= Post-rebuttal comment

After reading all reviews and responses, and discussing the common concerns with the other reviewers, one weak point of this work remains the increased capacity of the network (regardless of how the added parameters are optimized). For this reason, I chose to keep my original rating of WA.

**Time Spent Reviewing:**

4

---

> ### Author Response · Authors · 2021-08-10
> **Response to reviewer ab65**
>
> First, we would like to thank reviewer ab65 for the detailed review as well as for the insightful questions within.
>
> **Why is gradient cancellation a problem?** Gradient cancellation is considered a problem because it is a local issue that hampers training in MTL architectures. Gradient cancellation is tied with both parametrization and the optimization process. Finding direction conflicts between tasks during training does not necessarily mean that the tasks are not related, and thus the model cannot benefit from positive transfer. Often, local gradient conflicts lead to suboptimal optimization trajectories for all tasks and/or  to tiny gradient evaluations that prevent learning to occur. To see other nice illustrations of this issue refer, for example, to Figure 1 of PCGrad [33].
>
> **RotoGrad as increased capacity.** While one could initially think that RotoGrad has extra capacity in the right places, we would like to point out that  such an extra capacity is not optimized to minimize the task loss but instead RotoGrad’s own loss function (Eq. 3), i.e., to minimize gradient conflicts. To reinforce this key idea, we showed in Figure 4 that adding task-specific rotation matrices to IMTL-G (IMTL-G+$R_k$) optimized for improving the task loss, does not result in a competitive improvement in performance compared to "RotoGrad 512" (which leads to better performance with the same number of parameters). Thus, it is not only where to add the extra capacity, but how the additional parameters are optimized.
>
> **Noisy/lagging tasks.** We proposed a simple (hyperparameter-free) scaling algorithm that works well in all our experiments. However, such a simple algorithm would not work well in cases where tasks cannot be further learned due to limited network capacity or noisy labels. We will include a discussion on the limitations of the scaling algorithm, as well as potential future work to improve the scaling algorithm by, for example, learning a policy for better prioritizing tasks, in the revised version of the paper.
>
> **Replacing rotation matrices.** We are afraid that we do not immediately see how task-conditioned batch normalization could be applied to modify gradient directions (since a scalar multiplication can only modify the magnitude of the gradients of the individual tasks). We used rotation matrices since it is an intuitive approach when the goal is to rotate gradients. However, we are positive that RotoGrad can be further improved, and we hope that the core ideas presented in the paper are adopted and extended by the community.
>
> **Special treatment on RotoGrad’s gradient.** As discussed in L144-L152 and demonstrated in section 6.1, the Stackelberg formulation (only) needs the optimizer used for learning RotoGrad’s parameters to be a “slow-learner” compared with the one of the network’s parameters. This is however easily achievable through initial learning rate or scheduling.
>
> **Ablation study.** Please refer to the [general reply](https://openreview.net/forum?id=J_VRu0ZFVX-&noteId=RK8bbD33dpb).

---

### Official Review · Reviewer_wb8x · 2021-07-15

**Rating:** 6
**Confidence:** 4

**Summary:**

The paper proposes a method to reduce the optimization difficulty for multi-task learning. The key idea is to homogenize task gradients in terms of both magnitudes and directions. Specifically, they rescale task gradients to maintain similar convergence rates over tasks. At the same time, they introduce new network parameter R_k for each task k and train it so that the angle between task-specific and average gradients are minimized when task-head parameters are rotated using R_k. The method improves baselines and outperforms existing methods on DIGITS, CIFAR10, NYUv2, and CelebA datasets.

**Main Review:**

I think the key contribution of this paper is to introduce rotation matrices that reduces gradient conflicts by rotating the feature-space z in a way that matches their directions. In particular, the idea to use rotation matrices is reasonable and good in that they will preserve the relational information (e.g. distance) between features.

They claim that the novelty lies on the joint homogenization of gradients in terms of both magnitude and direction. However, IMTL [18] also jointly addressed loss magnitudes and gradient direction. Rather, I think novelty lies on the way they formulate the problem as a Stackelberg game. In other words, they optimize rotation matrices only to match gradient directions while optimizing the remaining network parameters to minimize the task loss. GradNorm used a similar strategy but they focused on adjusting gradient magnitudes.

In the experiments, their method improves baselines and performs better than other methods on four datasets and for two different backbone architectures. Additional analysis on training stability, effect of sub-dimension size for rotation, and training time comparison provides more comprehensive views.
However, it seems one needs to carefully adjust the training parameters for the rotation matrix, depending on dataset. when I ran the submitted code for mnist, the median cosine similarity for the task 'density' oscillated largely.


I think L130-131 is a bit misleading, because rotating the feature z is different from rotating its gradient g. Rotating z could result in the change of magnitudes in gradients. For example, consider a function f(x,y) = [relu(x),0]. A vector [x,y]=[1,0] and its rotation [x,y] = [-1,0] have different gradient magnitude.

It would be good to add ablation study to analyze the effect of magnitude and direction separately, i.e., magnitude-only and direction-only. Since each of them was explored in existing work, comparing each to corresponding other approaches will be also informative, e.g., magnitude-only to GradNorm and direction-only to PCGrad and GradDrop.

One way to demonstrate the training behavior will be to visualize the actual reduced gradient conflicts, possibly using angles, compared to the baseline. Also, comparing the method with baselines in the task loss curve will show that the method results in lower task loss.

**Time Spent Reviewing:**

7

---

> ### Author Response · Authors · 2021-08-10
> **Response to reviewer wb8x**
>
> We thank the reviewer for the great review and feedback.
>
> **Contributions.** We want to remark that homogenizing task gradients, both in magnitude and direction, is a key contribution of our work, as it is the first work tackling both problems simultaneously (see next paragraph). Yet, we agree with the reviewer that among the two components of RotoGrad, the rotations in Section 3.2 is the most novel one, and thus an important contribution. We strongly believe that both contributions will open new venues or research on MTL.
>
> **Prior work.** As mentioned by the reviewer, IMTL-G [18] is a direction-aware algorithm that focuses on scaling the gradients (not directly on homogenizing their directions). IMTL-G is a task-weighting approach—thus it can only change gradient magnitudes—whose goal is to scale each task gradient such that they are projected to the averaged one equally (thus direction-aware). In stark contrast, RotoGrad aims to homogenize the direction of the individual task gradients (i.e. make them equal in direction), which cannot be achieved by scalar multiplication of the gradients.
>
> While GradNorm [6] focuses only on the gradient magnitude, it indeed follows a similar strategy to ours, in the sense that it also includes per-task parameters that are optimized using an auxiliary loss (that in their case aims to balance gradient magnitudes across tasks). Remarkably, we have noticed that GradNorm can also be cast as a Stackelberg game. We will include this novel interpretation of GradNorm as a Stackelberg game in the appendix.
>
> **Tensorboard logs.** We apologize for the confusion regarding the tensorboard metrics. _In tensorboard, the metric of interest for RotoGrad is `cos_sim_{metric}`_, which computes the cosine similarity between task gradients and the average gradient. _The metric the reviewer refers to is a metric added during development just for the sake of debugging_, and which computes the cosine similarity for each dataset-instance and shows the median. Yet, we would like to stress that the oscillating behaviour observed by the reviewer is due to numerical instabilities, since gradients for `density` and `sum` in late-stages of training lie in the order of 1e-24.
>
> **Rotating z.** We thank the reviewer for the example and would like to confirm that the provided intuition is correct. Yet it slightly differs from what we mean. Using this same example, what we mean in L129-130 is that, since a rotation is bijective, we can substitute $f(x, y)$ by $f(R[x,y]) = (f \circ R)(x, y)$ in our optimization problem. As a consequence, the gradient of $f$ w.r.t to $R[x, y]$ (i.e., $r_k$ in the paper), is the same as that of $f \circ R$ w.r.t $[x, y]$ ($z$ in the paper), but rotated by $R$. This is stated in L138, and in the given example implies that $\nabla_{[x,y]} f(R[-1, 0]) = R^\top \nabla_{R[x,y]} f([1, 0])$. We will make sure to clarify this point  in the revised paper to avoid future similar misunderstandings.
>
> **Ablation study.** Please refer to the [general reply](https://openreview.net/forum?id=J_VRu0ZFVX-&noteId=RK8bbD33dpb).
>
> **Additional figures.**  We agree that these figures will easen the understanding of RotoGrad. As the reviewer saw, we already plot training information in tensorboard, and we will work to bring the figures of the cosine similarity and per-task losses during training to “Additional results” (Appendix C.2).

---

> > ### Comment · Reviewer_wb8x · 2021-08-25
> > **Reply to authors**
> >
> > I appreciate the authors for clarification and answers. My concerns are mostly addressed.
> >
> > I found that the implementation of GradNorm in the code looks different from the original paper. As far as I understand, G^{\bar}_W in the original paper is the average of gradient norms, whereas they implemented it as the norm of the averaged gradients (Line 23-24 in methods/gradnorm.py). I hope the authors can update the results of GradNorm in the revised version with proper search of alpha.

---

> > > ### Author Response · Authors · 2021-08-26
> > > **Re: Reply to authors**
> > >
> > > Dear Reviewer wb8x, we appreciate the feedback and we are happy to have addressed most of your concerns.
> > >
> > > Indeed, GradNorm uses as $\bar{G}$ the average of the norms, instead of the norm of the average. We apologize for the misstep. This has been corrected in our codebase and we have started re-running GradNorm's experiments, which we will definitely update in the next revision. Our apologies.
> > >
> > > We do still believe this should have minor effects on the results (and not affect the conclusions). This quantity, $\bar{G}$, is treated as the target value to homogenize all gradients' magnitudes (similar to $v$ in Eq. 3 of RotoGrad), so the algorithm should still be homogenizing the magnitudes. Since weights have to add up to 1, this mostly affects the learning rate to learn the weights, which we tune as described in L603-606 of the appendix.

---

### Author Response · Authors · 2021-08-10
**To all reviewers - ablation study**

We thank all reviewers for their invaluable time and feedback, as well as the general enthusiasm from all reviewers, which we really value and appreciate. Next, we address the main point raised by several of the reviewers, that is, the request of an ablation study to assess the contribution of each of the two components (gradient scaling and rotation) of RotoGrad.  To this end, we have extended the experiments on NYUv2 (Table 3) (as it is a heavy setup that involves tasks with completely different losses) to include:

1. Experiments where we run the two components of RotoGrad in isolation, that is:  i) our gradient scaling method (which we refer to as `ScaleOnly` in the table below) in Section 3.1 alone; and the gradient rotation algorithm (`RotateOnly`) in Section 3.2. With this experiment, we aim at demonstrating that both components of RotoGrad contribute to a more effective MTL approach.
2. Different versions of RotoGrad in which the gradient scaling component (in Section 3.1) is substituted by existing approaches, i.e., MGDA (leading to `RotoMGDA`), GradNorm (`RotoGradNorm`) and IMTL-G (`RotoIMTL-G`). With this experiment, we aim at evaluating the sensitivity of RotoGrad to its "magnitude" (i.e., gradient scaling) component.
3. Combinations of RotoGrad with complementary MTL methods to handle gradient conflicts, i.e., PCGrad and GradDrop. With this experiment, we aim to show that RotoGrad effectively handles conflicting gradient directions by itself.

A summary of the new results that will complete Table 3 in the revised paper is provided below. By looking at these new set of experiments, we can conclude that:
1. By jointly tackling both the magnitude and the direction conflicts, RotoGrad as proposed leads to better performance than when only one of its components is applied. That said, it is also clear that most of its benefit comes from the rotations (Section 3.2), which is in turn the main novelty of RotoGrad.
2. RotoMGDA, RotoIMTL-G, and RotoGradNorm clearly outperform their only scaling counterparts, leading to results that are comparable with the proposed RotoGrad but suffer in general from a higher computational complexity.  In conclusion, these results further demonstrate  the relevance of the rotation, as well as the effectiveness of the magnitude-scaling components of RotoGrad.
3. Combining PCGrad or GradDrop with RotoGrad does not lead to any improvements with respect to RotoGrad (rather on the contrary), which indicates that RotoGrad effectively addresses the gradient-direction conflicts.
As discussed already in the Conclusions, we plan to further analyze and extend RotoGrad (e.g., benchmarking) in future work. Yet,  we hope that the reviewers acknowledge these additional results, which we believe  further strengthen the novelty and contributions of the current paper.

Method | Semantic Segmentation $\operatorname{avg}_k\Delta_k$ | Depth  Estimation $\operatorname{avg}_k\Delta_k$ | Surface Normal $\operatorname{avg}_k\Delta_k$ | Hours
:-------:|:--:|:--:|:--:|:-----:
RotoGrad | 5.33 | 9.06 | -8.99 | 3.83
ScaleOnly | 5.26 | 4.79 | -8.64 | 3.51
RotateOnly | 3.93 | 7.53 | -8.40 | 3.84
RotoMGDA | 3.57 | 8.61 | -9.67 | 3.84
RotoGradNorm | 6.84 | 9.50 | -8.49 | 3.85
RotoIMTL-G | 4.25 | 4.07 | -8.94 | 4.30
RotoGrad + GradDrop | 5.40 | -6.92 | -10.14 | 3.96
RotoGrad + PCGrad | 2.83 | 7.86 | -8.85 | 3.91

---

> ### Comment · Area_Chair_YDvi · 2021-08-24
> **Discussion**
>
> @all_reviewers: Can you say whether the rebuttal answer your questions? Did this change your mind about the paper?
>
> Thank you

---

> ### Author Response · Authors · 2021-08-31
> **New experiments - Update**
>
> Dear reviewers and AC,
> on account of the reviewers feedback, we have kept running experiments that we want to share before the end of the discussion phase.
>
> Specifically, we have run most of the experiments in the main paper with the new implementation of GradNorm (we thank reviewer wb8x for the feedback), as well as on rotation-extended baselines, as it seems based on the updated reviews that model capacity is still a general concern.
>
> As before, we use: i) `method` + $R_k$ to denote a `method` that uses the architecture with rotations that minimize the tasks losses; ii) `Roto + method` to denote the combination of the direction part of `RotoGrad` with the scaling part of `method`; and iii) we denote by `GradNorm*` the new implementation of GradNorm, and by `GradNorm` the old one.
>
> **NYUv2**
>
> 1. Similar to IMTL-G, the updated GradNorm performs better on semantic segmentation and depth estimation, while worsening its results on surface normal.
> 2. Interestingly, the new GradNorm performs significantly worse when combined with RotoGrad. We hypothesize that this could be due to ignoring the direction of gradients to compute the norm ($||\sum_k g_k || \leq \sum_k ||g_k||$), so that RotoGrad gives bigger steps while still aligning gradients.
>
> |       Method        | Sem. Segm. $\operatorname{avg}_k\Delta_k$ | Depth  Est. $\operatorname{avg}_k\Delta_k$ | Surf. Normal $\operatorname{avg}_k\Delta_k$ | Hours |
> | :-----------------: | :--------------------------------------------------: | :----------------------------------------------: | :-------------------------------------------: | :---: |
> |    GradNorm     |                         -1.74                         |                       3.31                       |                     -5.25                     | 3.52  |
> |    GradNorm*    | 3.89 | 5.54 | -12.66 | 3.49 |
> | RotoGradNorm | 6.84 | 9.50 | -8.49 | 3.85 |
> | RotoGradNorm* | 3.50 | 6.47 | -10.29 | 3.84 |
>
> **CIFAR10**
>
> 1. Using extra capacity on vanilla does improve the results, yet it is far from RotoGrad's results with the same matrix sizes.
> 2. The updated GradNorm performs significantly better in this dataset compared with the older version.
> 3. The updated GradNorm + RotoGrad improves results, but it is still significantly worse that RotoGrad with the same matrix sizes.
>
> | Method          | $\operatorname{avg}_k\Delta_k$   | $\operatorname{med}_k\Delta_k$  | $\operatorname{max}_k\Delta_k$   |
> | --------------- | ----- | ---- | ----- |
> | Vanilla         | 2.58  | 2.73 | 11.14 |
> | Vanilla + $R_k$ | 3.11  | 3.09 | 14.22 |
> | GradNorm        | -0.08 | 0.09 | 8.82  |
> | GradNorm*       | 3.23  | 2.44 | 14.25 |
> | RotoGradNorm*   | 3.82  | 3.61 | 13.96 |
> | RotoGrad        | 4.48  | 4.72 | 15.57 |
>
>
> **CelebA**
>
> 1. The updated version of GradNorm performs better in the convolutional network, but worse in the residual one. Still, results are far from those from RotoGrad.
> 2. Vanilla with extended convolutional architecture improves all metrics, yet it is still far from RotoGrad's results.
> 3. In contrast, IMTL-G with extended ResNet18 architecture worses all metrics.
>
> |           | Conv. | Net   |       |       | Res.  | Net   | 18    |       |
> | --------- | ----- | ----- | ----- | ----- | ----- | ----- | ----- | ----- |
> | Method    |  $\operatorname{min}_k$   |  $\operatorname{med}_k$   |  $\operatorname{avg}_k$   |  $\operatorname{std}_k$   |  $\operatorname{min}_k$   |  $\operatorname{med}_k$   |  $\operatorname{avg}_k$   |  $\operatorname{std}_k$   |
> | Vanilla         | 1.62 | 54.74 | 58.69 | 24.18 |  15.45  | 61.51 | 61.24 | 22.09 |
> | Vanilla + $R_k$ | 4.93 | 58.55 | 60.26 | 23.02 | 18.35 | 62.49 | 63.03 | 21.16 |
> | GradNorm | 1.83  | 52.17 | 54.68 | 24.94 | 14.42 | 64.10 | 63.51 | 21.20 |
> | GradNorm* | 2.40  | 54.51 | 54.31 | 26.53 | 17.41 | 61.55 | 61.92 | 21.84 |
> | IMTL-G          |  3.31    |  53.04  | 56.04  | 26.91 | 21.52 | 62.12 | 61.98 | 21.62 |
> | IMTL-G + $R_k$  |  -  |   -    |  -   |   -    | 20.96 | 58.20 | 60.71 | 22.80 |
> | RotoGrad        | 9.11 | 62.31 | 62.45 | 22.14 | 25.72 | 63.84 | 65.17 | 18.99 |
>
> (IMTL-G + $R_k$ in the convolutional net ended mid-way through with a `CUDA error`, probably trying to invert a non-invertible matrix)
>
> ---
>
> **Remark.** We are aware that these new results are not complete, but we hope they are enough to help clarify the existing concerns. We are working hard to expand these _extra_ results before the end of the discussion phase and, in any case, we will make sure that they are fully complete by the time of the next revision.

---

### Decision · Program_Chairs · 2021-09-27

**Decision:**

Reject

**Comment:**

The work introduces an approach to minimize interference in multitask learning RotoGrad.

After carefully looking over the reviews and rebuttal, I do find the proposed solution interesting and the experimentation section comprehensive (especially with the additional results provided by the authors during the discussion period).
I do not feel that the concerns of extra parameters problematic in this scenario.

In general if there is a weakness of this work that I would highlight is that I feel more analysis could have been done to understand the behavior of the rotation matrices and the scaling of the updates done. For example, would the rotation matrices converge before the multitask problem converge to some fix task-specific rotation ? I would have been interested in any form of visualization that could have helped to understand what these matrices do, how they interfere with the baseline model, how they change the learned representation etc. The same for the scaling of the updates. I feel the focus is mostly on looking at the improved performance (which to be fair is the standard way of approaching this) but not as much on understanding how the proposed method works and what impact it has.

That said, the results are reasonable, and I do think this method will be of interest to the community.
After a long deliberation, I feel the paper in its current form is not exactly ready for NeurIPS (though is really a very borderline case). I think the additional results the authors provided greatly increase the paper. I would have liked a bit more analysis (or ablation) in understanding the interplay between the added capacity. What the rotation matrices really do. Anything that gives additional understanding of the dynamics of the method would have probably pushed the paper over. I strongly encourage the authors to improve the manuscript and resubmit/